# Comparative Study of Deep Transfer Learning Models for Semantic Segmentation of Human Mesenchymal Stem Cell Micrographs

**DOI:** 10.3390/ijms26052338

**Published:** 2025-03-06

**Authors:** Maksim Solopov, Elizaveta Chechekhina, Anna Kavelina, Gulnara Akopian, Viktor Turchin, Andrey Popandopulo, Dmitry Filimonov, Roman Ishchenko

**Affiliations:** 1V.K. Gusak Institute of Emergency and Reconstructive Surgery, 283045 Donetsk, Russia; mxsolopov@gmail.com (M.S.); neuro.dnmu@yandex.ru (D.F.);; 2Faculty of Medicine, Lomonosov Moscow State University, 119234 Moscow, Russia; voynovaes.pharm@gmail.com

**Keywords:** deep learning, mesenchymal stem cells, artificial intelligence, transfer learning, micrograph, phase-contrast microscopy

## Abstract

The aim of this study is to conduct a comparative assessment of the effectiveness of neural network models—U-Net, DeepLabV3+, SegNet and Mask R-CNN—for the semantic segmentation of micrographs of human mesenchymal stem cells (MSCs). A dataset of 320 cell micrographs annotated by cell biology experts was created. The models were trained using a transfer learning method based on ImageNet pre-trained weights. As a result, the U-Net model demonstrated the best segmentation accuracy according to the metrics of the Dice coefficient (0.876) and the Jaccard index (0.781). The DeepLabV3+ and Mask R-CNN models also showed high performance, although slightly lower than U-Net, while SegNet exhibited the least accurate results. The obtained data indicate that the U-Net model is the most suitable for automating the segmentation of MSC micrographs and can be recommended for use in biomedical laboratories to streamline the routine analysis of cell cultures.

## 1. Introduction

Mesenchymal stem cells (MSCs) are attracting increasing attention from researchers in the field of regenerative medicine due to their wide range of isolation sources, potential to differentiate into various tissue types, and pronounced immunomodulatory and anti-inflammatory effects [1]. These properties make MSCs a promising therapeutic agent for the treatment of various diseases, especially infectious diseases [2] and musculoskeletal [3] and neurological disorders [4]. A necessary step in the study of MSCs is a careful analysis of cell morphology [5] and their ability to proliferate and differentiate [6], alongside the quantification of cell culture parameters (e.g., contact surface area, cell volume) [7]. Traditionally, light microscopy methods have been used to monitor MSCs, allowing researchers to visually assess morphological characteristics, adhesion, and colony formation. However, this approach is time-consuming, requires human resources, and reduces the reproducibility of results due to subjective interpretations by the operator. With the development of artificial intelligence (AI) techniques, particularly deep learning using convolutional neural networks, it has become possible to automate and accelerate the process of analyzing cell culture micrographs [8,9].

One way of applying neural network models for analyzing cell cultures is through semantic image segmentation (SIS). This method can automatically identify and isolate individual cells or their clusters in micrographs by separating them from the background. For this purpose, a neural network model is trained on a large number of images in which the regions of cell locations are manually annotated by experts in cell biology. During training, these models automatically extract image features while preserving and restoring spatial information due to the encoder–decoder architecture of the neural network. The encoder is responsible for successively reducing the spatial resolution of the image using convolution (highlighting key features such as textures and shapes) and pooling (reducing the image size to decrease computational complexity), allowing the model to automatically extract and generalize features of cellular structures. This process is similar to information compression, where the model learns to recognize key features of cells at different levels of abstraction. The decoder, in turn, performs the inverse task of restoring the original resolution of the image by upsampling (increasing resolution to restore details) and convolution (to refine boundaries and key features) to accurately represent the boundaries and details of the segmented cells. The decoder often takes information from the encoder layers at corresponding levels and directly transmits it to the appropriate decoder layers through special pass-through connections. This preserves fine image details, which improves segmentation accuracy since the model uses both the generalized features and the initial spatial data. Based on this processed data, the model classifies each pixel of the image to a specific category, such as “cell” or “background”, enabling the precise identification of distinct regions. SIS data from cell culture micrographs can be used to assess morphological phenotypes [10], intracellular structures [11] and cell area [12].

The manual annotation of cell culture micrographs by experts to create large-scale datasets suitable for machine learning is a time-consuming and labor-intensive task. Consequently, the number of publicly available annotated datasets of human cell culture micrographs is small. The EVICAN [13] and LIVECell [14] datasets, for instance, contain 4600 and 5239 annotated micrographs, respectively. One promising approach to optimize the workload is the use of transfer learning. This method allows for the use of models that have been pre-trained on large sets of heterogeneous images and adapting them to segment cell micrographs. The pre-trained models are effective at recognizing basic patterns and structures, and additional training on small datasets of cell micrographs eventually allows them to achieve competitive segmentation accuracy [15].

Although there are published works demonstrating the successful application of transfer learning using neural network architectures such as U-Net [16], DeepLab [15], and Mask R-CNN [17] for computer vision in cell culture analysis, comparative data on which algorithms are best suited for the segmentation of MSC micrographs are still insufficient. To address this gap, we conducted a comparative evaluation of several popular neural network models—U-Net, DeepLab, SegNet and Mask R-CNN—for the segmentation of human MSC micrographs. The selected architectures were chosen due to their widespread use in biomedical segmentation, their ability to effectively learn from limited datasets (through transfer learning), and their complementary advantages: U-Net provides baseline accuracy, DeepLabV3+ enhances detail through multi-scale analysis, Mask R-CNN isolates individual objects, and SegNet combines simplicity with computational efficiency. We aim to determine which of the models has the best segmentation performance after training on a dataset of 320 cell micrographs using the transfer learning method, which will allow us to make recommendations for their application in local biomedical laboratories for training on their own micrographs. To this end, we established an expert-annotated dataset of MSC micrographs with ground truth masks and conducted rigorous model training and evaluation against expert-derived masks.

## 2. Results

### 2.1. Training Evaluation of Segmentation Models

Figure 1 shows the dynamics of pixel accuracy (PA) and loss function changes for the considered models in the learning process. All models demonstrate a stable decrease in the loss function, reaching stable PA values at the final epochs of training. There are no signs of overfitting—a situation where training accuracy improves while validation accuracy declines.

Notably, the plots of PA dynamics reveal different trends across the models. The U-Net and DeepLabV3+ models exhibit a smooth and gradual increase in PA as training progresses, indicating stable optimization and adaptation to the training data. In contrast, the SegNet model displays oscillatory changes in PA over a wide range before stabilizing after 40 epochs. A systematic improvement in PA for the Mask R-CNN model was observed after 60 epochs. The time spent on training each model was as follows: (1) U-Net—21 min 55 s; (2) DeepLabV3+—16 min 18 s; (3) SegNet—16 min 01 s; and (4) Mask R-CNN—18 min 38 s. These durations reflect the computational demands associated with each model’s complexity.

### 2.2. Analysis of Model Performances Using Optimized Thresholds

To improve the accuracy of the segmentation masks, we investigated the effect of different prediction thresholds on the quality metrics of the models (see Figure 2, which illustrates the relationship between prediction thresholds and quality metrics). For all models, we calculated the average values of the metrics over a validation sample for each threshold and determined the optimal threshold that maximizes the numerical values. As a result of the analysis, the following optimal thresholds were determined: for the U-Net model, the optimal thresholds for the Dice coefficient (DC), Jaccard index (JI) and PA metrics were 0.43, 0.43, and 0.50, respectively; for DeepLab, the optimal thresholds were 0.30, 0.30, and 0.42; for SegNet, the optimal thresholds were 0.15, 0.15, and 0.37; and for Mask R-CNN, the optimal thresholds were 0.05, 0.06, and 0.17. The similarity in optimal thresholds for the DC and JI metrics is due to the mathematical relationship between the two metrics. Despite the differences in their calculation formulas, both metrics assess similar aspects of overlap between the predicted mask and the ground truth segmentation mask.

Table 1 shows the results of the metric calculations using the optimal threshold for maximizing the DC and JI. The value for maximizing the best DC was chosen as the threshold for mask prediction. The results of the Friedman test demonstrated significant differences between the models for all three evaluated metrics (*p* < 0.05). A Dunn’s post hoc test was then performed to identify the best-performing model, and the results are presented in Figure 3.

The U-Net model shows the best results in terms of the DC metric, reaching an average value of about 0.876. Statistical analysis showed that the differences between U-Net and other models are significant, but to different degrees. U-Net has the greatest superiority compared to SegNet (*p* < 0.0001) and to a lesser extent compared to DeepLabV3+ (*p* < 0.01) and Mask R-CNN (*p* < 0.05). The DeepLabV3+ and Mask R-CNN models also demonstrated competitive DC values, with no statistically significant differences between them in terms of the DC metric. Additionally, these models, as well as U-Net, significantly outperform SegNet (*p* < 0.0001). A similar interaction pattern between the models was observed in the calculation of JI metrics. The U-Net model demonstrated the highest value at 0.781, while the DeepLabV3+, SegNet, and Mask R-CNN models showed the highest values of 0.749, 0.600, and 0.750, respectively.

The analysis of model performance in terms of PA showed that there were no statistically significant differences between the U-Net, DeepLabV3+ and Mask R-CNN models. The PA scores were quite high for all three models: 0.935, 0.922, and 0.923, respectively. The SegNet model showed a much lower score of 0.858.

### 2.3. Visual Evaluation of Segmentation Performance

For a visual evaluation of the performance of the investigated neural network models, several examples of original micrographs from the validation sample, along with ground truth masks and predicted masks, are presented in Figure 4, which illustrates a comparison between actual and predicted segmentations. The U-Net, DeepLabV3+, and Mask R-CNN models demonstrated the highest accuracy in extracting cell boundaries, particularly when compared to the SegNet model, which performed less effectively in extracting complex and small structures of cell morphology. On the masks predicted by SegNet, flaws in segmentation such as rough boundaries and missing details can be observed, especially in areas where cells have curved or complex shapes. Despite the high segmentation accuracy of the Mask R-CNN model, the masks predicted by this model are characterized by poorly defined cell edges. These irregularities appear as jagged contours. This may be due to the fact that Mask R-CNN, focusing on the instance-segmentation task, selects objects of certain classes in the image. The algorithm focuses on the exact localization of objects but does not always reproduce boundaries. In this respect, U-Net and DeepLabV3+ demonstrate greater accuracy in contour reconstruction.

## 3. Discussion

In this study, we conducted a comparative evaluation of the performance of the U-Net, DeepLabV3+, SegNet, and Mask R-CNN neural network models for the semantic image segmentation (SIS) of MSC micrographs. The segmentation accuracy of the models was validated against expert-annotated ground truth masks, which served as the benchmarking standard. The primary metrics to evaluate the segmentation quality were the Dice coefficient (DC) and Jaccard index (JI). These metrics are considered the most reliable for SIS tasks, as they evaluate the degree of agreement between ground truth and predicted masks, considering both correctly selected regions and omissions or false selections. Unlike pixel accuracy (PA), which measures the proportion of correctly classified pixels and can be skewed by class imbalance, the DC and JI are more sensitive to the quality of object segmentation. In the context of MSC micrograph segmentation, where cells may occupy a relatively small image area compared to the background, PA can be a misleading metric. A high PA value can be achieved even if the model does not segment cells well but correctly classifies many background pixels. Therefore, the DC and JI are more reliable metrics for assessing model performance, as they focus on the accuracy of cell boundary and shape detection.

The results showed that the U-Net model achieved the best segmentation accuracy in terms of the DC metric, significantly outperforming the other models. U-Net also demonstrated high pixel accuracy comparable to the DeepLabV3+ and Mask R-CNN models. The high performance of U-Net can be attributed to its architectural features, particularly its U-shaped structure with skip connections between the encoder and decoder, which preserves both low- and high-level image features. This is crucial in biomedical image segmentation, where fine details and cell contours must be accurately reproduced. Our findings are consistent with previous studies in which U-Net has been successfully applied to the segmentation of other cell types [16,18].

The DeepLabV3+ model also showed high results. Its architecture allows it to account for objects of different scales and shapes, which is useful when working with cells of diverse morphology. However, the lack of direct gaps between layers may result in the loss of some spatial details, likely explaining the slight lag in metrics compared to U-Net. Adnan et al. applied DeepLab models for the segmentation of MSC micrographs but faced limitations due to a small and partially annotated dataset, as well as low JI metrics [15].

Mask R-CNN demonstrated high metric values but showed shortcomings in cell boundary segmentation accuracy, as detected during a visual analysis of the predicted masks. The insufficiently accurate extraction of edges on the predicted masks may be related to its focus on instance-based segmentation, which prioritizes selecting individual objects but does not always provide high accuracy in outline extraction. In cell segmentation, where accurate boundary delineation is important, this may be a drawback for the practical application of this model.

The least effective model in our study was SegNet, which failed to qualitatively recognize cell morphology patterns due to its optimized architecture aimed at reducing computational cost. SegNet lacks direct connections between the encoder and decoder, limiting the transmission of low-level features. This can lead to the loss of important spatial information and degrade the model’s ability to distinguish small and complex structures, as reflected in lower segmentation metric values.

The use of pre-trained models and the transfer learning method allowed us to achieve high performance even with a relatively small dataset. This highlights the effectiveness of transfer learning for segmentation tasks in the biomedical research field, consistent with other studies [19,20]. However, our dataset consisted of only 320 images, which may be a limitation for the generalizability of the results. Increasing dataset size and a variety of micrograph acquisition conditions could improve model performance.

The manual annotation of micrographs is a subjective process and can introduce additional variability into the data. Using semi-automated annotation methods involving multiple experts to create consensus masks can enhance the quality of the reference data.

The training time of the models was comparable and was not a significant factor in selecting the optimal model. Thus, U-Net is the most suitable tool for segmenting micrographs of human MSCs.

Our results have practical implications for biomedical laboratories, which can use their own trained U-Net-based models to automate the routine analysis of cell culture micrographs. Laboratories can train the U-Net model on their own small datasets, creating algorithms customized to their characteristics. This will reduce the time for image processing and increase the objectivity and reproducibility of results when assessing the morphological characteristics of MSCs and other cell lines of similar morphology. Additionally, these models can monitor cell culture confluency, which is crucial for determining the optimal subculture timing. This opens up opportunities for integrating U-Net models into robotic systems [21], allowing for the automation of cell subculturing and accelerating research in regenerative medicine.

## 4. Materials and Methods

### 4.1. Dataset and Image Annotation

A dataset of micrographs of human MSCs obtained from five independent donors was generated for this study. Cell cultures were grown under standard conditions in a CO_2_-incubator: 37 °C, 5% CO_2_, 95% humidity, and DMEM/F12 nutrient medium (Thermo Fisher Scientific, Waltham, MA, USA) supplemented with 10% fetal calf serum (HyClone, Logan, UT, USA) and 1% penicillin/streptomycin (HyClone, Logan, UT, USA). Micrographs were obtained during routine cell culture monitoring using a Nikon Eclipse Ti2 (Nikon Instruments Inc., Melville, NY, USA) inverted light microscope equipped with a 10× magnification lens in phase-contrast mode.

A total of 320 micrographs, each measuring 1000 × 1000 pixels, were collected. To create a set of ground truth masks, all images were manually annotated by three independent experts. The annotation process involved highlighting the areas corresponding to cells and creating binary masks, in which the pixels of cells were assigned a value of 1 and the background was assigned a value of 0. For this purpose, the VGG Image Annotator v3.0.11 was used, which provides tools for the accurate marking of complex objects in images.

### 4.2. Neural Network Models for Segmentation

Four neural network models for image segmentation—U-Net, DeepLabV3+, SegNet, and Mask R-CNN—were selected and implemented with Python programming language v3.10 using the open-source library TensorFlow v2.15.0. These architectures are widely used in academic and commercial institutions under Apache 2.0 or MIT licenses, which allow for extensive reuse and modification [22,23]. Their broad applicability, particularly in medical imaging, stems from their ability to balance performance, computational efficiency, and adaptability to diverse tasks.

U-Net, originally designed for biomedical segmentation, remains a benchmark model in medical imaging due to its encoder–decoder architecture with skip connections, which preserves spatial details while capturing both local and global features. A pre-trained VGG16 model acts as the encoder. U-Net has been extensively validated across applications such as tumor detection, cell tracking, and organ segmentation, establishing it as a robust baseline for biomedical tasks [24]. However, its large number of trainable parameters can increase overfitting risks on small datasets.

DeepLabV3+ addresses complex segmentation challenges by employing atrous (dilated) convolutions and an Atrous Spatial Pyramid Pooling (ASPP) module, which enables multi-scale contextual understanding without drastically increasing computational complexity. This design allows the model to segment objects of varying sizes in medical images, such as lesions or anatomical structures at different resolutions [22]. The integrated Xception model as an encoder further enhances its ability to capture fine details through hierarchical feature extraction.

SegNet, another encoder–decoder architecture, prioritizes computational efficiency by reducing memory usage through indexed pooling operations. With fewer parameters than U-Net, it is particularly suitable for scenarios with limited labeled data or hardware constraints, making it ideal for real-time applications [22].

Mask R-CNN, an extension of Faster R-CNN, adds a parallel branch for instance segmentation, enabling pixel-level mask prediction for individual objects. This capability is critical for distinguishing overlapping or adjacent regions in medical images, such as clustered cells or intersecting organs [22]. Utilizing ResNet-50 and a Feature Pyramid Network (FPN) as the encoder, the model excels in object-level differentiation while maintaining strong detection accuracy across scales.

### 4.3. Model Training Procedure

Neural network models were trained on the Kaggle web platform using NVIDIA Tesla T4 GPU. Each of the selected models was pre-initialized with weights obtained after training on the ImageNet dataset. This approach, known as transfer learning, leverages features already trained on a large set of images and accelerates training on a new task, improving accuracy and preventing overfitting, especially with limited data.

The dataset was divided into training and validation samples in a ratio of 80:20; 256 images were used to train the models and 64 images were used to evaluate their performance. Image augmentation techniques (random reflections, rotations, and rescaling) were applied to increase the diversity of the dataset and prevent overtraining.

The models were trained over 100 epochs. One epoch is a process in which the model goes through the entire training set of images once, adjusting its parameters to improve prediction accuracy. Binary cross-entropy was used as the loss function, considering the binary nature of the segmentation task (cell/background). Model parameters were optimized using the Adam optimizer with standard settings. To monitor the learning process, loss function and pixel accuracy values were tracked as quality metrics on training and validation samples.

### 4.4. Statistical Data Processing

Groups for comparing the performance of segmentation models were formed based on the same set of 64 validation images. The metrics described in Table 2 were calculated for each model. The values of all metrics range from 0 to 1, with 1 corresponding to an ideal match between the predicted and reference segmentations.

For each model, an optimal numerical threshold was determined. This threshold is used to convert the probability predicted by the neural network for each pixel into a binary value (0 or 1). It is essential to divide the image into areas of cells and background using this threshold. The threshold was varied from 0 to 1 in increments of 0.01. For each threshold value, the average values above the specified metrics were calculated for a validation sample of images.

To identify statistically significant differences between models, the nonparametric Friedman test for related samples was applied since all models were evaluated on the same set of images. The normality of the data distribution in the groups was preliminarily assessed using the Shapiro–Wilk test. Levene’s test was used to check the equality of variance between the groups. For further analysis, Dunn’s post hoc test with Bonferroni correction for multiple comparisons was used. This allowed us to identify which pairs of models have statistically significant differences in terms of the estimated metrics. All statistical calculations were conducted using the Python libraries SciPy v1.14.1 for the basic tests and scikit-posthocs v0.10.0 for the post hoc analysis.

## 5. Conclusions

The application of transfer learning enabled the efficient training of neural network models on a small dataset, confirming the feasibility of this approach for segmenting MSC micrographs. In a comparative analysis of the efficiency of the neural network models U-Net, DeepLabV3+, SegNet, and Mask R-CNN for the SIS of cell culture micrographs, the U-Net model achieved the best segmentation accuracy according to the DC and JI metrics, surpassing the other models.

The results support the recommendation of the U-Net model for use in biomedical laboratories to automate MSC micrograph segmentation. This will accelerate cell culture analysis and enhance the reproducibility and objectivity of the results. Future research plans include expanding the micrograph dataset, exploring other neural network architectures, and investigating the impact of additional data augmentation and regularization methods on model accuracy.

## Figures and Tables

**Figure 1 ijms-26-02338-f001:**
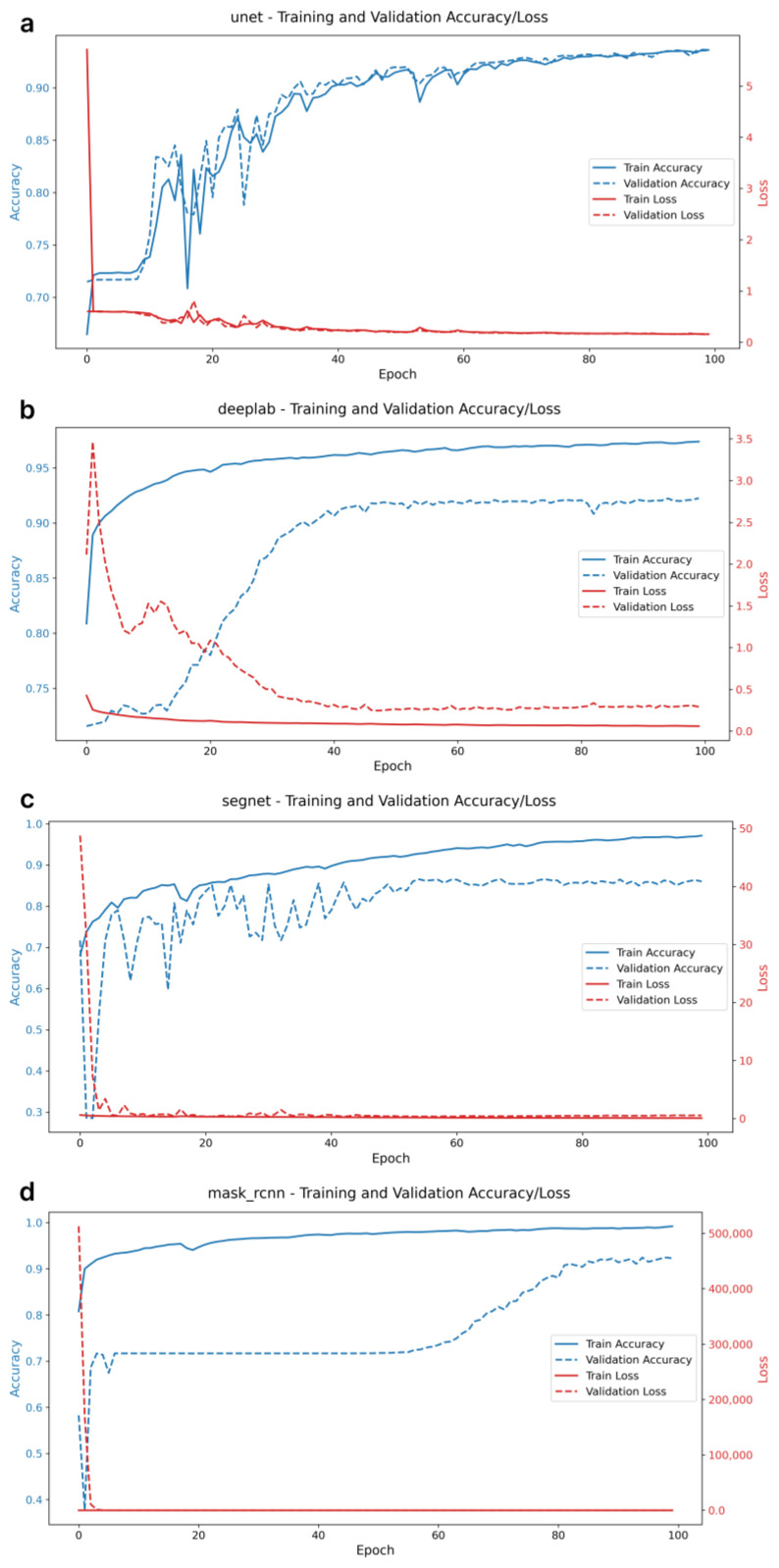
Training graphs of neural network models for segmenting micrographs of mesenchymal stem cells (MSCs). Dynamics of changes in pixel accuracy (PA) and loss function for the investigated models on training and validation samples of micrographs during training: (**a**) U-Net, (**b**) DeepLabV3+, (**c**) SegNet, and (**d**) Mask R-CNN.

**Figure 2 ijms-26-02338-f002:**
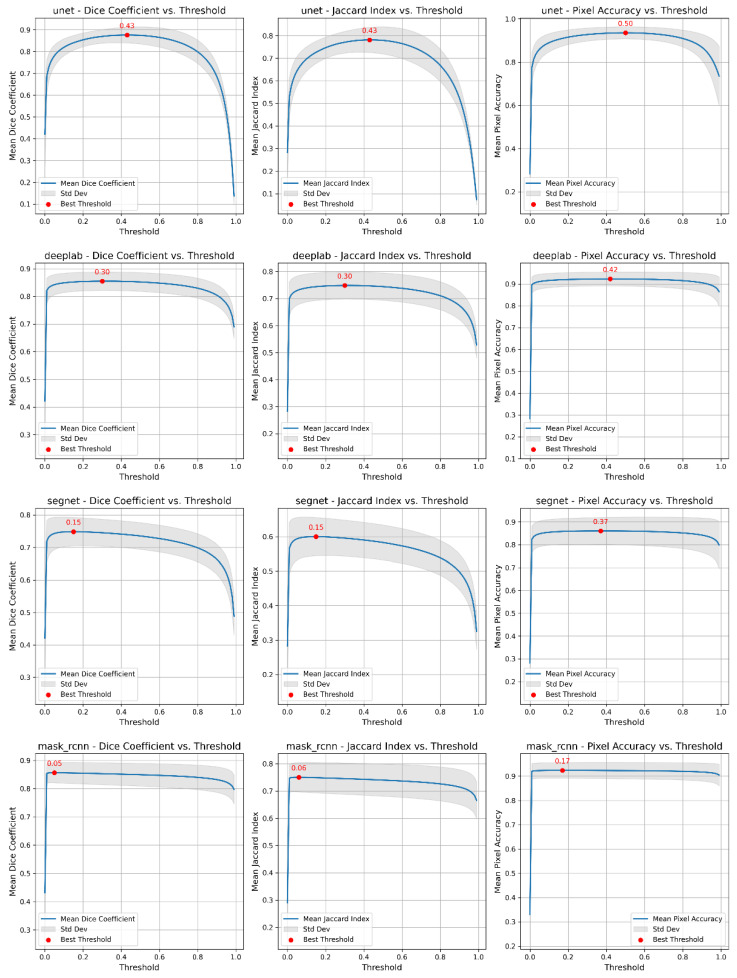
Optimal prediction thresholds for U-Net, DeepLabV3+, SegNet, and Mask R-CNN segmentation models (from top to bottom) according to the Dice coefficient (DC), Jaccard index (JI) and PA metrics (from left to right). The optimal thresholds are defined as the maximum values of the functional dependencies of the metric on the threshold value. To plot the dependencies, the average value of each metric was calculated over 64 images from the validation sample at a given value of the varying threshold. The graphs show the mean values (blue line) with standard deviations (highlighted in gray).

**Figure 3 ijms-26-02338-f003:**
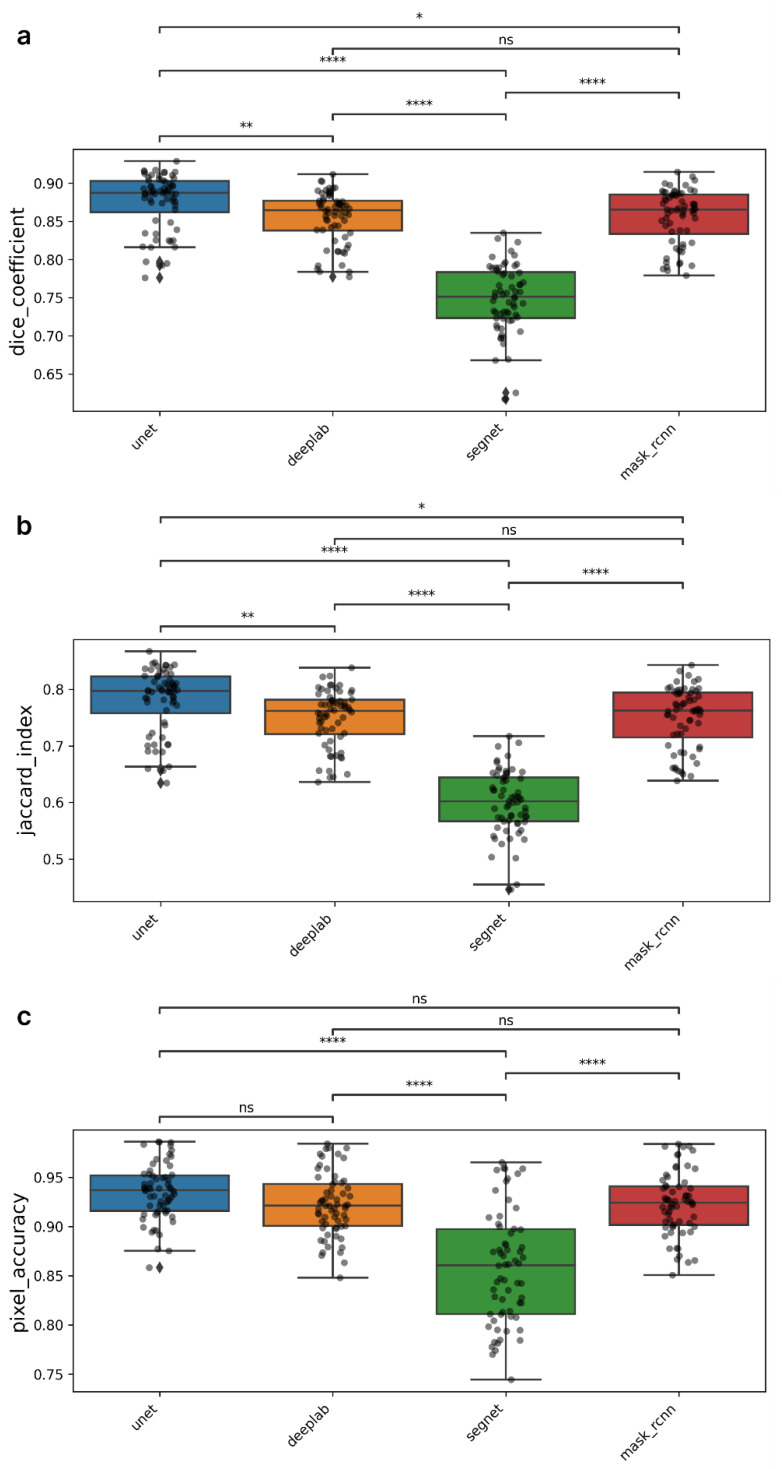
Comparison of the performance of segmentation models based on DC (**a**), JI (**b**), and PA (**c**) metrics. The charts show the distribution of metric values for each model. * *p* < 0.05; ** *p* < 0.01; **** *p* < 0.0001; ns—differences are not significant.

**Figure 4 ijms-26-02338-f004:**
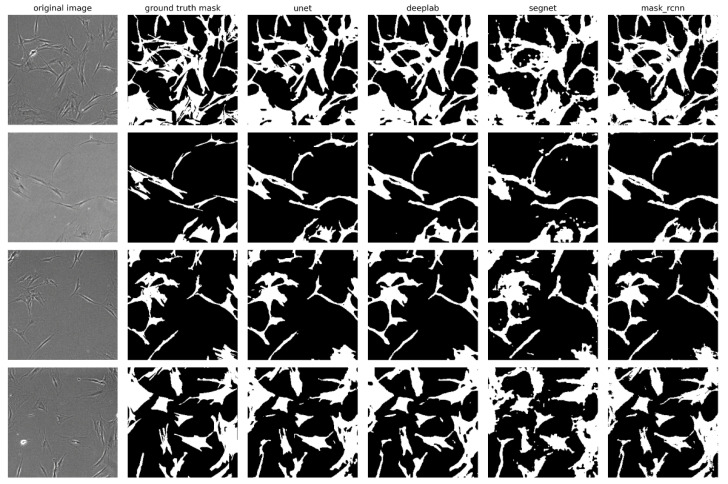
Examples of segmentation of MSC micrographs by neural network models: original images, ground truth masks, and masks predicted by U-Net, DeepLabV3+, SegNet, and Mask R-CNN models. The micrographs were captured at a magnification of 40×.

**Table 1 ijms-26-02338-t001:** Metrics of trained neural network models for SIS of MSC micrographs.

Model	DC	JI	PA	Threshold
U-Net	0.876 ± 0.036	0.781 ± 0.058	0.935 ± 0.028	0.43
DeepLabV3+	0.855 ± 0.033	0.749 ± 0.050	0.922 ± 0.031	0.30
SegNet	0.748 ± 0.044	0.600 ± 0.055	0.858 ± 0.057	0.15
Mask R-CNN	0.856 ± 0.036	0.750 ± 0.053	0.923 ± 0.032	0.05

Notes. The mean values calculated for the validation sample of images (n = 64) and the values of the standard deviations are given for DC, JI, and PA. For all metrics, a value of 1 denotes perfect overlap with the expert-annotated ground truth masks.

**Table 2 ijms-26-02338-t002:** Metrics used to evaluate the performance of neural network models for SIS of MSC micrographs.

Title	Formula	Description
Pixel accuracy (PA)	$\frac{\sum 1(y_{true}=y_{pred})}{\sum 1(y_{true}\geq 0)}$	Proportion of correctly classified pixels.
Dice coefficient (DC)	$\frac{2\cdot \sum \left(y_{true}\cdot y_{pred}\right)}{\sum y_{true}+\sum y_{pred}}$	The degree of overlap between the predicted mask and the ground truth mask considering both accuracy and completeness of segmentation.
Jaccard index (JI)	$\frac{\sum \left(y_{true}\cdot y_{pred}\right)}{\sum y_{true}+\sum y_{pred}-\sum \left(y_{true}\cdot y_{pred}\right)}$	The similarity between the predicted and ground truth mask is defined as the ratio of their intersection area to their union area.

Notes. *y_true_* is the true (reference) label, which contains values of 1 for pixels belonging to the cell in the micrograph and 0 for the background; *y_pred_* is the label predicted by the model.

## Data Availability

The dataset is available at https://www.kaggle.com/datasets/maximsolopov/msu-smooth-1-20 (accessed on 20 January 2025). The notebook with statistical calculations is available at https://www.kaggle.com/code/maximsolopov/segmentation-of-mscs-images (accessed on 20 January 2025).

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
