# Peer review of "Comparative Study of Deep Transfer Learning Models for Semantic Segmentation of Human Mesenchymal Stem Cell Micrographs"

_ijms, 2025, doi:10.3390/ijms26052338_

Round 1
Reviewer 1 Report
Comments and Suggestions for Authors
Dear Authors,
the analysis of microscopic images of live cell cultures on ideal cell culture surfaces, especially for stem cells and primary, differentiated cells, is highly dependent on many factors. Even with standardised laboratory protocols in the form of SOPs and under GLP conditions, it also depends on the experience of the respective personnel. The time factor for the evaluation is also considerable. Today, it is imperative to implement modern automation and AI applications.
Your comparative study of ‘deep transfer learning models for semantic segmentation’, here of hMSCs, is enormously beneficial for the global life sciences community for an international harmonisation of image data evaluations and their standardisation.
This work is very well and clearly designed, explained in a largely plausible manner, and the results are well suited for applications to more complex, morphologically inhomogeneous cell types. The authors' conclusion regarding the usability of the U-net model is consistent with our own experiences.
There are a few notes from me for improvement, which I have added as comments to the uploaded version.

Reviewer 2 Report
Comments and Suggestions for Authors
- The authors need to provide details on the source and type of the human MSC used in this study. The authors used 20% of FBS for MSC culture, which is a high percentage. Any reason?
- The authors applied four neural network models for image segmentation—U-Net, DeepLabV3+, Mask R- 265 CNN and SegNet. The rationale of selecting those platforms needs to be clearly explained. Is there any gold standard method to be used as a control?
- The authors need to provide cell pictures and conventional methods for analyzing cell phenotype in comparison to their platforms.
Need to improve the quality of English and check the grammatical mistakes and typos.
